

# Potential contribution of fish restocking to the recovery of deteriorated coral reefs: an alternative restoration method?

Uri Obolski[1], Lilach Hadany[1] and Avigdor Abelson[2]

[1] Department of Molecular Biology and Ecology of plants, Tel Aviv University, Tel Aviv, Israel
[2] Department of Zoology, Tel Aviv University, Tel Aviv, Israel

## ABSTRACT

Counteracting the worldwide trend of coral reef degeneration is a major challenge for the scientific community. A crucial management approach to minimizing stress effects on healthy reefs and helping the recovery of disturbed reefs is reef protection. However, the current rapid decline of the world's reefs suggests that protection might be insufficient as a viable stand-alone management approach for some reefs. We thus suggest that the ecological restoration of coral reefs (CRR) should be considered as a valid component of coral reef management, in addition to protection, if the applied method is economically applicable and scalable. This theoretical study examines the potential applicability and outcomes of restocking grazers as a restoration tool for coral reef recovery—a tool that has not been applied so far in reef restoration projects. We studied the effect of restocking grazing fish as a restoration method using a mathematical model of degrading reefs, and analyzed the financial outcomes of the restocking intervention. The results suggest that applying this restoration method, in addition to protection, can facilitate reef recovery. Moreover, our analysis suggests that the restocking approach almost always becomes profitable within several years. Considering the relatively low cost of this restoration approach and the feasibility of mass production of herbivorous fish, we suggest that this approach should be considered and examined as an additional viable restoration tool for coral reefs.

Corresponding author
Avigdor Abelson, avab23@gmail.com, avigdor@tauex.tau.ac.il

## INTRODUCTION

Coral reefs are considered to be among the most threatened and fastest deteriorating marine ecosystems (*Burke et al., 2011*; *Knowlton & Jackson, 2008*). At present, a well-accepted approach to countermeasure reef decline is that of 'conservation,' which focuses on the protection of reefs from overuse and misuse (e.g., over-fishing and destructive fishing), and the removal of local stressors, if such exist (*Hughes et al., 2010*; *Mumby & Steneck, 2008*). Nonetheless, coral reef degeneration has remained a major challenge for the scientific community (*Hughes et al., 2010*), and the present conservation-based approach seems insufficient to serve as a stand-alone solution.

An alternative approach, aimed at targeting this challenge, is coral reef restoration (CRR; also termed coral reef rehabilitation *Edwards & Gomez, 2007*; *Rinkevich, 2005*). The

common 'CRR approach' posits promoting the recovery of reefs mainly through coral reef gardening: the transplantation of stony corals, much in the way that nursery stock is planted in terrestrial gardens (*Edwards, 2010*; *Edwards & Gomez, 2007*; *Rinkevich, 2005*). At present, however, CRR remains a subject of controversy in the coral reef research community. The major arguments against CRR include its limited scalability (*Adger et al., 2005*; *Mumby & Steneck, 2008*); the ineffectiveness of restoration efforts in the face of natural threats, such as climate change and ocean acidification (*De'ath, Lough & Fabricius, 2009*; *Mumby & Steneck, 2008*; *Pandolfi et al., 2003*); and the high costs of the prevailing CRR approaches, i.e., reef gardening and artificial reefs (*Adger et al., 2005*; *Mumby & Steneck, 2008*). Much of the criticism of the restoration approach stems from the view that CRR, in its present state, is practically limited to a single method, i.e., coral reef gardening, which is currently attracting the major efforts of restoration interventions and scientific research (*Edwards, 2010*; *Edwards & Gomez, 2007*; *Rinkevich, 2008*).

In the present study we propose the approach of restocking grazing fish as an additional CRR method, and examine its possible efficiency and economic value. Restocking (also termed re-introduction or biomanipulation of fish populations; *Angeler, 2010*; *Cowx, 1999*) is a common tool in the applied management of non-marine aquatic ecosystems, aimed at restoring water quality and vegetation characteristics (*Angeler, 2010*; *Cowx, 1999*; *Cowx & Gerdeaux, 2004*). Although less used in the marine environment, restocking has recently been applied to coastal marine ecosystems, mainly as a fishery management tool aimed at recovering the yields of target commercial fish populations (*Leber, 2013*; *Lindegren, Mollmann & Hansson, 2010*; *Lorenzen et al., 2013*; *Lorenzen, Leber & Blankenship, 2010*). Moreover, there have been some attempts at restocking in coral reefs, mostly of invertebrate species (e.g., the grazing gastropod *Trochus sp.*; *Castell, Naviti & Nguyen, 1996*; *Villanueva, Edwards & Bell, 2010*), but also fish stock enhancement (e.g., rabbitfish and parrotfish; *Bowling, 2014*).

Restocking of grazing fish in coral reefs is based on the following rationale: most degraded reefs undergo a phase-shift from coral-dominated reefs to algal-dominated ones (mostly macroalgae, or algal turfs). Such degraded reefs are likely to remain in their unfavorable state if not inhabited by enough grazers. Since the natural recovery of grazing fish is very likely to take years (or even decades; *Blackwood, Hastings & Mumby, 2012*), stock enhancement of key grazing species is expected to significantly accelerate the process. Given that stock enhancement has been successful in other marine systems (e.g., kelp forests and rocky coastal habitats), and that the technologies for culturing some species of grazing fish already exist (*Bowling, 2014*; *Duray, 1998*), a restoration approach based on stock enhancement seems to be worth examination.

To examine the possible ecological outcomes and economic feasibility of restocking grazing fish as a potential restoration tool for degraded reefs, we: (1) applied reef dynamic models (*Blackwood, Hastings & Mumby, 2011*; *Blackwood, Hastings & Mumby, 2012*; *Mumby, Hastings & Edwards, 2007*) to compare recovery rates under various conditions of conservation and restoration; and (2) performed a cost-benefit analysis to compare the financial implications of restocking over time according to the model; that is to determine

whether some of the limited funds available for reef conservation should be allocated to restoration or rather solely to conservation.

## METHODS

### The model

We examine the potential outcomes of fish restocking using modifications of mathematical models, consisting of differential equations, which have been used to examine reef resilience without intervention (*Blackwood, Hastings & Mumby, 2011*; *Blackwood, Hastings & Mumby, 2012*; *Fung, Seymour & Johnson, 2011*; *Mumby, Hastings & Edwards, 2007*).

The dynamic model we use here has been adapted from *Blackwood, Hastings & Mumby (2011)*, the most recent version of the model first presented in *Mumby, Hastings & Edwards (2007)*. This model enables us to follow the dynamics of coral coverage, macroalgae, algal turfs, grazing fish, and terrain rugosity; denoted, respectively, by the variables $C, M, T, P$ and $R$. The algal turf coverage refers to the proportion of seabed either covered by scant and small algae, such as might occur due to grazing of macroalge, or exposed hard substrate. We assume that the corals, macroalgae, and algal turfs are competing for seabed in a constant size location, and define the algal turf coverage to be *1-M-C*. The variable $P$ describes the abundance of grazing fish relative to the maximum capacity of grazing fish possible in the habitat, so that $0 < P < 1$. Rugosity is defined as the ratio between the horizontal distance of two points in the reef and the length of a chain laid on the reef surface between those points, usually satisfying $1 < R < 3$ (*Alvarez-Filip et al., 2009*). The rest of the dynamics are given by the following set of ordinary differential equations:

$$\frac{dM}{dt} = aMC - \frac{g(P)M}{M+T} + \gamma MT$$
$$\frac{dC}{dt} = rTC - dC - aMC$$
$$\frac{dT}{dt} = \frac{g(P)M}{M+T} - \gamma MT - rTC + dC \tag{E1}$$
$$\frac{dP}{dt} = sP\left(1 - \frac{P}{K(M+T,R)}\right) - mP - fP$$
$$\frac{dR}{dt} = h_G C(3-R) - H_e(1-C)(R-1).$$

We assume that corals grow on a seabed covered with algal turfs at rate $r$, die of natural causes at rate $d$, and are covered by macroalgae at rate $a$. Macroalgae too grow over algal turfs, at rate $\gamma$. Grazing fish grow according to the logistic growth equation, with growth rate $s$, and a maximal carrying capacity function $K(M+T, R)$, which is the product of a linear, increasing, function of rugosity and a Hill-Langmuir function of $M+T$ (for details see *Blackwood, Hastings & Mumby, 2011*). The grazing fish graze macroalgae into algal turfs at a rate $\frac{g(P)M}{M+T}$, where $g(P)$ is taken to be $P$. Thus, the grazing fish reduce the rate of macroalgae growth over corals, while simultaneously expanding algal turfs, which can provide seabed for coral growth (*Jompa & McCook, 2002*). Grazing fish are fished at rate $f$.

**Table 1   Coral growth model parameters.**

| Parameter | Value | Meaning |
|---|---|---|
| $a$ | 0.1 | Rate of macroalgae overgrowth on corals |
| $\gamma$ | 0.8 | Rate of macroalgae overgrowth on algal turfs |
| $r$ | 1 | Rate of coral growth on algal turfs |
| $d$ | 0.44 | Natural coral mortality |
| $f$ | 0 | Fishing rate of grazing fish (we assume fishing restrictions) |
| $s$ | 0.49 | Grazing fish growth rate |
| $h_G, h_E$ | 0.03 | Growth and erosion rates of reef complexity, respectively |
| $m$ | 0.12 | Spillover rate |
| $\delta_p$ | 0.1 | Proportion of grazing fish restocked, normalized to the carrying capacity |

Terrain rugosity increases due to growth of corals, $h_G$, and decreases due to bioerosion at a rate $h_E$, where both $h_G$ and $h_E$ depend on current rugosity and coral coverage (functions were estimated from data by *Blackwood, Hastings & Mumby, 2011*). Grazing fish migrate from the reef at rate $m$. The parameter values and their meanings are given at Table 1. All parameter values used are taken from *Blackwood, Hastings & Mumby (2011)*, except for spillover estimates ($m$). We used spillover rates estimated in *Kaunda-Arara & Rose (2004)* (acquired by the tag and release method) and converted them to yearly migration rates assuming an exponential decrease, in order for them to fit the parametrization of our equations (see Text S1). We first consider the effect of restocking on a single reef, and then extend the model to two reefs, each represented by the system of differential equations presented above. The reefs are coupled by the migration parameter. We assume that all fish from one reef, denoted Reef I, migrate to another reef, denoted Reef II; while fish from Reef II migrate as in the one reef system, and are effectively lost in our model. This is a conservative assumption, as we examine the worst case in terms of restocking benefit in which none of the fish from Reef II migrate to Reef I. Additionally, we assume that both reefs are relatively close, so that climatic or anthropogenic perturbations will affect both reefs similarly and bring them to the same initial conditions (a distance of $\sim$10 km might be an estimate for such conditions; *Hughes et al., 1999*). We introduce restocking by adding an amount $\delta_P$ to the initial value of $P$, namely $P_0$. Since $P$ is normalized to be between 0 (no grazers) and 1 (maximal capacity of grazers), $\delta_P$ is given as a fraction of the maximal abundance of grazing fish possible in the modeled habitat.

We model the economic impact of fish restocking using a cost-benefit analysis. We define $X$ as the size of the coral reef in km$^2$; $B_C$ is the revenue, per km$^2$ per year, resulting from coral coverage (excluding revenue from fishing); $\hat{K}$ is the maximal carrying capacity of the restocked grazer fish per km$^2$, and $r'$ is the economic discount rate. For $t$ years, the difference in revenue for a coral reef with restocking versus a reef with no intervention can be estimated by (also known as the net present value):

$$\text{Revenue}(t) = X \cdot B_C \cdot \sum_{i=0}^{t} (1+r')^{-i} (C^{\text{re}}(i) - C^{\text{no}}(i)) - (\tilde{c} \cdot \hat{K} \cdot X \cdot \delta_p) \qquad (\text{E2})$$

**Table 2  Economic model parameters.**

| Parameter | Value | Meaning |
|---|---|---|
| $X$ | 15 | Coral reef size (km$^2$) |
| $B_C$ | 200,000 | Benefit of coral reef $\left(\frac{\text{USD}}{\text{year} * \text{km}^2}\right)$ |
| $r'$ | 0.05 | Discount rate |
| $\hat{K}$ | 3,000 | Estimated grazing fish carrying capacity $\left(\frac{1}{\text{km}^2}\right)$ |
| $\tilde{c}$ | 50 | Estimated cost per fish (USD) |

where $C^{re}(i)$ and $C^{no}(i)$ are the coral coverages of reefs with and without restocking, at year $i$, respectively. Simply put, we calculate the difference in coral coverage between a reef with and without the restocking intervention. This difference is multiplied by the size of the reef and the financial benefit for each squared kilometer of the reef. The term is discounted with regard to inflation. Finally, the cost of restocking is subtracted. This is a conservative estimate, since in this model restocking increases coral coverage, and it is assumed that the benefit from coral reefs declines relative to the cost of the fish, due to discounting. The parameters and variables of the economic model describing the revenue are given in Table 2. The Matlab code for all results is given in File S3.

## RESULTS

### A single reef

First, we examine the long-term effects of restocking for various initial conditions of the dynamic system presented above, with a single coral reef. We assume that as part of the restoration treatments, fishing restrictions are implemented, so that $f = 0$ in all the following results. The state of a disturbed reef is represented by the initial conditions of the coral coverage $(C)$ and macroalgae $(M)$ (determining the amount of algal turfs, as $T = 1 - M - C$). The revenue of restocking (derived from (E2)) as well as the final outcomes of the restocking intervention (derived from (E1)), are presented as functions of the system's initial conditions in Fig. 1.

Since this dynamic system has two attractors, one of high coral coverage and the other of high macroalgae coverage (*Blackwood, Hastings & Mumby, 2011*), the range of initial conditions can be divided into 3 areas: (I) initial conditions in which the system reaches a state with high coral coverage with or without restocking; (II) areas wherein the system would reach a high macroalgae state in the absence of intervention (but under fishing restrictions), but restocking would allow its return to the high coral coverage state; and (III) areas where the system will reach a state with high macroalgae coverage with or without restocking. These areas are denoted in Fig. 1 as (I), (II) and (III), respectively, and are separated by black borders. In addition, colors in Fig. 1 represent the expected revenue of restocking, 5 and 20 years after restocking has taken place, in $\log_{10}$ scale, with negative revenue replaced by zeros (Figs. 1A and 1B). Note that the variables are normalized to represent the entire reef area, so that $T = 1 - M - C$ and the state of the algal turfs $(T)$
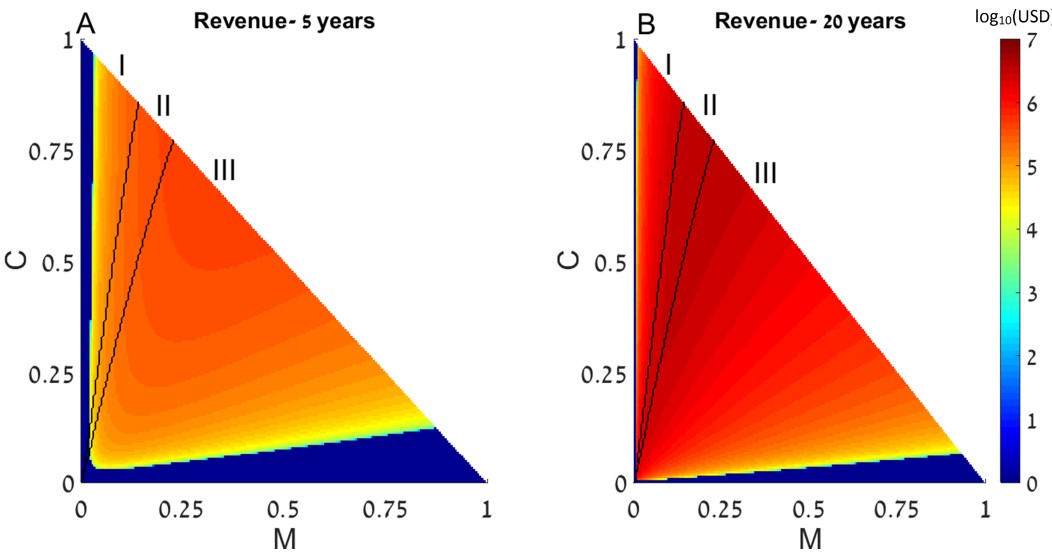

**Figure 1** **Revenue of restocking.** The expected revenue of restocking a 15 km² reef is represented by a color scale (in $\log_{10}$ scale), as a function of the reef's initial macroalgae and coral coverage (horizontal and vertical axes, respectively). Note that $T = 1 - M - C$ and the state of the algal turfs ($T$) is defined by the other two variables. The revenue is calculated for 5 years (A) and 20 years (B) after restocking has been implemented. Areas in which the revenue is negative are replaced by zeros on the color scale (note that the negative revenue is bounded from below by the initial cost of restocking). Black curves divide the plot into three initial condition areas: (I) the system reaches a state with high coral coverage with or without re-stocking; (II) the system reaches a high macroalgae state in the absence of intervention but high coverage under restocking; and (III) the system reaches a state with high macroalgae coverage with or without restocking. Parameter values are given in the main text.

is defined by the other two variables. The parameters used in Fig. 1 for the dynamical system were $P_0 = 0.1, \delta_P = 0.1, R_0 = 1.6$ (the rest of the dynamical system parameters were given the values estimated in *Blackwood, Hastings & Mumby, 2011*). The spillover was estimated from odds of tagged fish leaving and staying in the coral reefs (*Kaunda-Arara & Rose, 2004*), and was transformed to a rate term to yield $m = 0.12$. The reef size ($X$) was taken to be 15 km², the estimated grazing fish number per km² ($\hat{K}$) was taken as 3,000, estimated from *Gaudian, Medley & Ormond (1996)*, according to the density of the most common fish in the examined coral reef (accounting for 64% of all fish). Thus, when we enhance the number of grazing fish by $\delta_P = 0.1$, we de facto add 300 fish per each km² of reef area. The financial benefit from the coral reef ($B_C$) was estimated from *Cesar & Van Beukering (2004)*, as $200,000 per year per km², which is a very conservative estimate (see *Spurgeon, 1999*, for example). The average cost of each fish, $\tilde{c}$, was estimated to be $20 as an over-estimated price. This estimated cost is based on the recent average fish price for cultured fish (ca. $1.8/kg) taken from fish price trends in real terms during the last two decades (*FAO, 2014*). Estimating an average size of 500 gr of released fish results in a cost of $0.9 per fish. The actual fish cost, however, should be calculated based on the expected survival rates of the released fish. Estimating a survival rate of at least 10% of the released fish (*Hervas et al., 2010*), implies a release of ca.10 times the size of the desired population size. Therefore, the realistic (yet over-estimated) cost should be set at $10 per fish, and to

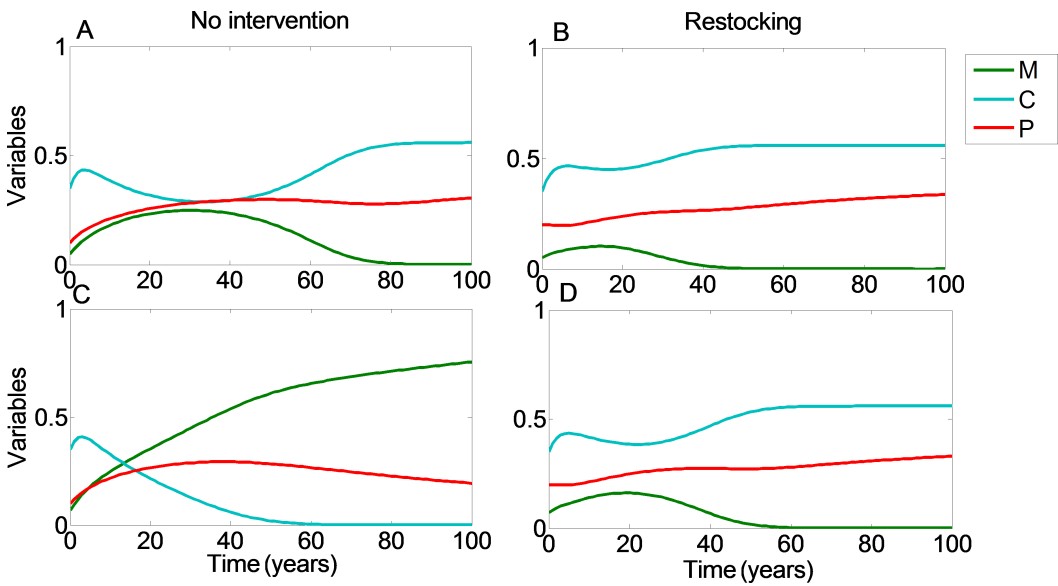

**Figure 2  Restocking shortens restoration time.** We plot the values of the coral coverage ($C$, light blue), macroalgae coverage ($M$, green), and grazing fish ($P$, red) with respect to time, no intervention (A and C), and restocking (B and D), for different initial conditions. (A, B) present the model variables simulated from initial values in which the reef will be restored without intervention ($C_0 = 0.35, M_0 = 0.05$). (C, D) represent initial conditions in which the coral reef will deteriorate without intervention, but will return to high coral coverage when restocking is implemented ($C_0 = 0.35, M_0 = 0.07$). Other parameter values are as in Fig. 1.

account for variance of estimates we have multiplied this by a factor of 2. Therefore, using our estimates, the cost of restocking a reef spanning 15 km$^2$ will amount to approximately $\tilde{c} \cdot \hat{K} \cdot X \cdot \delta_p = 20 \cdot 3{,}000 \cdot 15 \cdot 0.1 = 90{,}000$ USD.

The discount rate ($r'$) was set to 0.05, but our results remain robust when varying discount rates (Text S2).

From Fig. 1 we can see that restocking broadens the range of conditions under which the system will reach a high coral coverage state, but not to a very substantial extent. However, restocking increases the expected revenue of a disturbed reef under a wide range of initial conditions, especially in the long term (Fig. 1, compare A to B). This is the result of the relatively cheap cost of restocking (estimated at 6,000 USD per km$^2$), combined with the high revenue of coral reef area (estimated at \$200,000 per year per km$^2$). Even if the reef does not restore to high coral coverage, the delay in its deterioration, enabled by restocking, will still be profitable for a large extent of the initial conditions. Similarly, even if the reef will eventually be restored without human intervention, restocking will shorten the period of time required to achieve this. This is shown in Fig. 2, where a time series of the values of the coral coverage ($C$), macroalgae coverage ($M$), and grazing fish ($P$) are plotted with and without restocking for two sets of initial conditions. Figures 2A and 2B presents the model variables simulated from initial values corresponding to area (I) in Fig. 1 ($C_0 = 0.35, M_0 = 0.05$), in which the reef will be restored without intervention. Although both scenarios lead to an eventual high coral coverage, we can see that restocking will shorten the time to equilibrium to about 65% of this time in a system without restocking

(compare Figs. 2A and 2B). A change in initial conditions, to those corresponding to area II in Fig. 1, can change the dynamics entirely. Figures 2C and 2D represents initial conditions ($C_0 = 0.35, M_0 = 0.07$) in which the coral reef will deteriorate without intervention (Fig. 2C), but will return to high coral coverage when restocking is implemented (Fig. 2D).

In area III, the reef would remain in a high macroalgae state with or without restocking. In such a case, we could consider a combination of restoration methods. For instance, if feasible, eradication of macroalgae will be expressed as moving left on the phase plane presented in Fig. 1 in our model. We expect that when restocking is applied, the extent of eradication needed to bring the system to a point where restoration will be possible will be lower, and the restoration time from that point will be shorter.

## Multiple reefs

We next generalize the notion of restocking to a system consisting of two reefs, in which the fish migrate from one reef to another. We define the direction of migration from Reef I (upstream) to Reef II (downstream). Under this range of initial conditions we note five possible scenarios: (I) initial conditions under which in both reefs the system reaches a state with high coral coverage without restocking; (II) areas wherein one reef will reach high coral coverage without intervention, while the other reef will only succeed if restocking is applied; (III) areas wherein one reef will reach high coral coverage without intervention, while the other will reach the macroalgae state even if restocking is applied; (IV) areas in which both reefs will deteriorate to the macroalgae state without intervention, but restocking will salvage one of them; and (V) areas in which both rates will deteriorate and restocking will not help either reef. These areas are marked accordingly on Fig. 3. Additionally, colors in Fig. 3 represent the expected revenue of restocking 5 and 20 years after the restocking has taken place, in $\log_{10}$ scale, with negative revenue replaced by zeros. Parameters of Fig. 3 are as in Fig. 1, with the migration from Reef I is directed towards Reef II, and migration from Reef II is lost.

We can see that restocking broadens the range of conditions under which at least one of the reefs will reach a high coral coverage state. Moreover, restocking increases the expected revenue from the coral reefs under almost all conditions. This is due to the amplification of the effect seen in Figs. 1 and 2. Even when restocking is only performed for one of the reefs, it accelerates the return to a high coral coverage state, and delays deterioration of the reefs. Figure 4 presents time-series examples for these dynamics for the same parameters as in Fig. 2. $C_I, M_I, P_I$ and $C_{II}, M_{II}, P_{II}$, are the coral coverage, macroalgae coverage and grazing fish, for the upstream and downstream reefs, respectively. We can see that restocking only the upstream reef can shorten the recovery time to about 60, for both the downstream and upstream reefs %, relative to the system without restocking (Fig. 4 compare A to B). In addition, for parameters that are within region II of Fig. 3, restocking can salvage the upstream reef from deterioration (Fig. 4 compare C to D).

## DISCUSSION

Studies carried out in the last decade suggest that the protection of coral reefs as MPAs (Marine Protected Areas) is a useful tool for the maintenance of coral cover (*Selig & Bruno,*

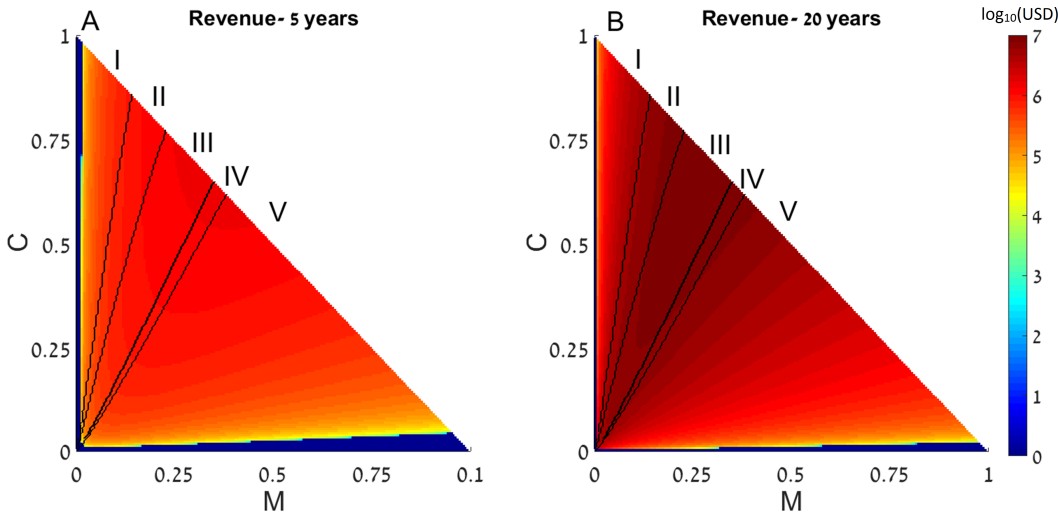

**Figure 3  Revenue of restocking in two connected reefs.** The expected revenue of restocking in one reef, connected by fish migration to another reef, is represented by a color scale (in $\log_{10}$ scale), as a function of the reefs' initial macroalgae and coral coverage (horizontal and vertical axes, respectively). Note that $T = 1 - M - C$ and the state of the algal turfs ($T$) is defined by the other two variables. The revenue is calculated for 5 years (A) and 20 years (B) after restocking has been implemented. Areas in which the revenue is negative are replaced by zeros on the color scale (note that the negative revenue is bounded from below by the initial cost of restocking). Black curves divide the plot into five areas according to initial conditions leading to different outcomes. The areas are marked by Roman numerals, and explained in the main text. Parameters are as in Fig. 1, with migrating fish from the restocked reef ending up in the other reef.

2010), reef resilience, and recovery (*Mumby & Harborne, 2010*). However, most coral reefs around the world have not been protected. Moreover, many coral reefs are not in an optimal healthy state due to diverse stressors, mainly anthropogenic: e.g., over-fishing, habitat destruction, pollution, and climate-change related effects (*Burke et al., 2011*; *De'ath et al., 2012*). The question thus arises as to what should be the appropriate management approaches in those numerous coral reefs that have already become significantly degraded. A key concern is whether *ad hock* protection can serve as a stand-alone tool to help in the natural recovery of these reefs, or might it not suffice (*Huntington, Karnauskas & Lirman, 2011*). In the latter case, additional management approaches might be required to enable improvement of the reefs' state and to prevent further deterioration.

The general notion that proactive human intervention will be critical for mankind's survival, health, and prosperity, is becoming increasingly common among terrestrial ecology scientists and decision-makers (*Dobson, Bradshaw & Baker, 1997*; *Suding, 2011*). In contrast, the mainstream scientific approach does not consider restoration as an applicable management tool for coral reef ecosystems (e.g., *Adger et al., 2005*; *Mumby & Steneck, 2008*; but conversely see *Abelson et al., 2016*; *Rinkevich, 2014*).

In this work we used a mathematical model to examine the feasibility and potential efficiency of fish population restocking, aimed at accelerating coral reef recovery. The proposed 'restocking' tool, as applied to fishery enhancement management, is based on previous efforts to enhance wild fish populations by releasing cultured fish into aquatic environments (*Leber, 2013*). Ideally, fish from the local population would be used as the

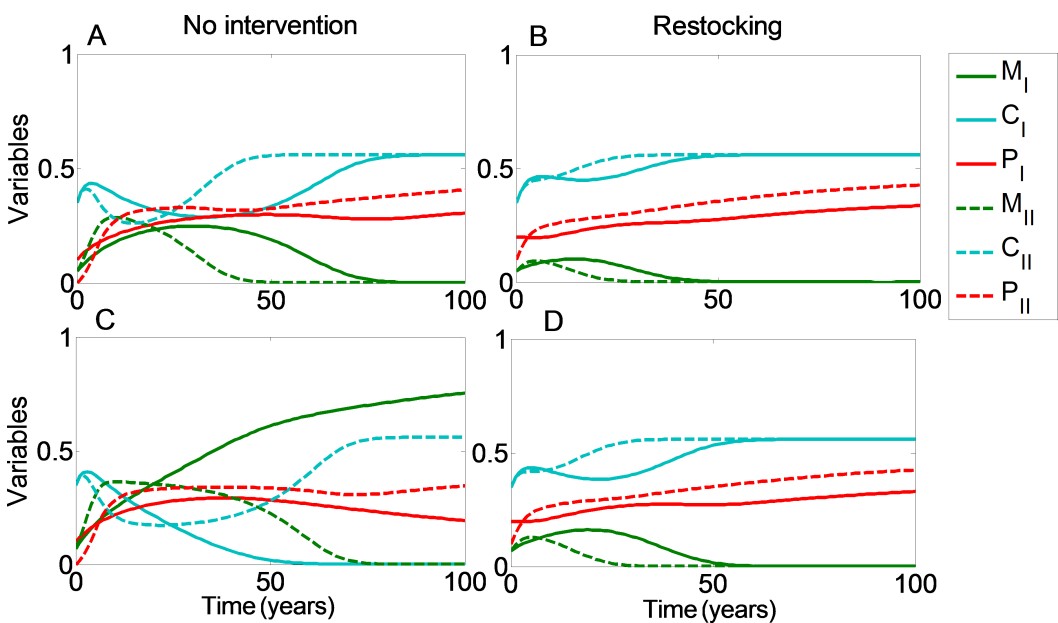

**Figure 4 Restocking shortens restoration time in connected reefs.** The model's variables the ($C$, coral coverage; $M$, macroalgae coverage; $P$, grazing fish) are plotted with respect to time, under no intervention (A and C) and restocking (B and D) for two reefs connected by migration. $C_I, M_I, P_I$ and $C_{II}, M_{II}, P_{II}$, represent the coral coverage (light blue), macroalgae coverage (green), and grazing fish (red), for the upstream (dashed lines) and downstream (solid lines) reefs, respectively. (A, B) present the model variables simulated from initial values in which both reefs will be restored without intervention ($C_0 = 0.35, M_0 = 0.05$). (C, D) represent initial conditions in which the upstream coral reef will deteriorate without intervention, but will return to high coral coverage when restocking is implemented ($C_0 = 0.35, M_0 = 0.07$). Other parameter values are as in Fig. 3.

brood of the cultured fish for restocking, and the brood population would be large enough to limit the loss of variability due to founder's effect (*Champagnon et al., 2012*).

However, the restocking tool is not suggested as a management solution for recovery of every degraded reef. It has been shown that beyond $0.5B_0$ (where $B_0$ is the average biomass of resident reef fish in the absence of fishing; *MacNeil et al., 2015*) fishery restrictions can in themselves be successful in sustaining key functions of reef fish such as herbivory (*MacNeil et al., 2015*). Thus, we suggest that the 'restocking' restoration solution be examined in severely depleted sites, such as heavily fished reefs (e.g., reefs in Jamaica, Guam and Papua New Guinea; *Knowlton & Jackson, 2008*; *MacNeil et al., 2015*), which also guarantees that reintroducing fish into the reef will not harm the homeostasis of the ecological system, but rather contribute to restoring it.

The dynamics of the grazers selected for restocking should satisfy several conditions. The carrying capacity of grazers should depend on the amount of coral coverage, and increase with increased coverage. In contrast, coral coverage cannot be so high that the macroalgal coverage would be insufficient to support the feeding needs of the grazers. However, because the coral coverage does not tend to exceed the threshold of food limitation for grazers (*Blackwood, Hastings & Mumby, 2011*), this is not a substantial limiting factor of the model's generality. To maintain the grazers at substantial quantities within the perimeter

of the reef, both fishing and migration rates of the grazers should not be high. While our analysis assumed complete fishing restrictions and intermediate migration rates, similar results would be obtained with a low amount of fishing permitted and low migration, since the fishing and migration parameters work in the same manner in the model (see 'Methods' and Supplemental Information 1). Finally, the grazers must exert a grazing pressure that is sufficient to produce a significant effect on the macroalgal coverage. Some of the grazers that fulfill the above assumptions are certain Parrotfish genera (*Mumby et al., 2006*; *Williams & Polunin, 2001*) and siganid fish (*Siganus virgatu*) (*Plass-Johnson et al., 2015*), which also seems to be a feasible taxon for culturing (*Duray, 1998*).

Importantly, our results show that restocking is a financially beneficial method, due to the high economic value of coral reef services (*Caillaud et al., 2011*; *Cesar & Van Beukering, 2004*) and the potentially low cost of restocking (*Lorenzen et al., 2013*). In addition, fish restocking has the advantage that it does not require full-cover intervention of the entire reef area. Such restocking is intended to be applied in spatially-limited focal spots, which will subsequently serve as potential rehabilitation hotspots for further (natural) recovery of the rest of the reef area, via spillover of adult grazers, or by larval supply from the restored patches as sources of 'flourishing populations' (*Abesamis & Russ, 2005*; *Selig & Bruno, 2010*). Thus when the reef is clearly in a more severe state, we should consider implementing additional interventions concurrently with restocking. For instance, in dense macroalgae-dominated reefs, restocking can be ineffective, as fish tend to remain outside dense algal forests (*Hoey & Bellwood, 2011*). Furthermore, some grazing fish can alter their main source of nutrition in response to changes in the abundance of algae types (*Khait et al., 2013*). If, on the other hand, macroalgae eradication alone is applied, given that future research will indeed show that this is a cost-effective method of restoration, the reef is expected quickly to become covered again by macroalgae due to the lack of grazers (*McClanahan et al., 2000*). In such a situation, restocking following macroalgae eradication can promote natural recruitment. Such combined interventions might prove to have synergistic interactions, and to be even more efficient and economically beneficial. Another possible intervention is that of coral transplantation, also termed reef gardening, in which corals are directly planted into a reef (*Edwards, 2010*; *Rinkevich, 2005*). Although this method directly increases the coral coverage and the reef's structural complexity (rugosity), it is estimated at about 200,000–1,300,000 USD per $km^2$ for low-cost transplantations (*Edwards & Gomez, 2007*). Therefore, when comparing between the two alternative restoration tools, under the circumstances discussed above, even if the reef gardening method is highly effective, the relatively negligible cost of restocking (estimated here at 6,000 USD per $km^2$), and its potential benefit should at least incentivize the implementation of both tools concomitantly.

It is our expectation that future research will yield further ecological and economic estimates, which could help us to assess the efficiency of such interventions and of their combinations.

It should be stressed that our proposed restoration approach is not presented as an alternative to protection. Moreover, we agree with the widely-accepted notion that protection (including removal of stressors, if applicable) is the most important management

tool by which to maintain reef health and to facilitate the fast recovery of reefs following wide-scale natural disturbances. We propose, nonetheless, that fish restocking, and possibly other ecological restoration tools in conjunction with conservations measures, be considered as an efficient and economically beneficial method for the rehabilitation coral reefs.

### Funding

This project was supported by the Israel's Ministry of Environmental Protection (AA), supported in part by the Israeli Science Foundation 1568/13 (LH), and by a fellowship from the Manna Program in Food Safety and Security (UO). The funders had no role in study design, data collection and analysis, decision to publish, or preparation of the manuscript.

### Grant Disclosures

The following grant information was disclosed by the authors:
Israel's Ministry of Environmental Protection.
Israeli Science Foundation: 1568/13.
Manna Program in Food Safety and Security.

### Competing Interests

The authors declare there are no competing interests.

### Author Contributions

- Uri Obolski conceived and designed the experiments, performed the experiments, analyzed the data, wrote the paper, prepared figures and/or tables, reviewed drafts of the paper.
- Lilach Hadany and Avigdor Abelson conceived and designed the experiments, analyzed the data, wrote the paper, reviewed drafts of the paper.

### Data Availability

  The research in this article did not generate any raw data.

### Supplemental Information

Supplemental information for this article can be found online at http://dx.doi.org/10.7717/peerj.1732#supplemental-information.

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
