# Peer review of "Potential contribution of fish restocking to the recovery of deteriorated coral reefs: an alternative restoration method?"

_PeerJ, doi:10.7717/peerj.1732_

## Round 0.1 · original submission · Major Revisions

Apologies for the delay with this decision, which was caused by the need to find a 2nd reviewer after someone dropped out.

Please consider all the suggestions in the revised manuscript.

Reviewer 1 ·

Basic reporting

In the manuscript number #2015:09:6663:0:0:REVIEW, entitled "Potential contribution of fish restocking to the recovery of deteriorated reefs: an alternative restoration method?" Uri Obolski, Lilach Hadany and Avigdor Abelson model the effect of an intervention in degraded coral reefs with the increase of the population of herbivorous fishes using cultivation. They argue that faced the actual degradation condition of coral reefs, using solely an approach of conservation biology is not enough; they advocate an increase in research and then use of restoration techniques on these ecosystems. This is a very interesting manuscript, but there is a major problem that must to be solved before this paper should be considered ready to publish: the discussion focuses entirely on the need of a restoration approach, but without discussing their own results. I agree that their results are very clearly related to their discussion, but point out that nonetheless we could remove their methods and results sections entirely without any change to their discussion argument logic and the meaning of that part of the text. In this regard, it becomes an opinion or a point of view paper. In addition, there are some concerns which need more attention to address, stated below.

Experimental design

This manuscript show a mathematical model and not a experiment, so there is no experimental design. However, the authors begin their model separating the macroalgae from turf algae. But, in a later step, both are considered the same. These two groups have different effects on coral populations and on the reef ecosystem. There are several recent publications that report a phase shift to macroalgae as well as turf algae (Jouffray et al 2015, Naumann et al. 2015 and Gove et al 2015). Macroalgae are more aggressive in their interactions with coral populations; turf algae, in contrast, inhibit coral recruits, and also, in several cases, could be involved in the spread of coral disease. Furthermore, herbivorous fishes are more efficient at controlling turf algae than macroalgae. I recommend that the authors do separate model components for macroalgae dominance and turf algae dominance.

Validity of the findings

This is a very interesting manuscript, but there is a major problem that must to be solved before this paper should be considered ready to publish: the discussion focuses entirely on the need of a restoration approach, but without discussing their own results. I agree that their results are very clearly related to their discussion, but point out that nonetheless we could remove their methods and results sections entirely without any change to their discussion argument logic and the meaning of that part of the text. In this regard, it becomes an opinion or a point of view paper. In addition, there are some concerns which need more attention to address, stated below.

Additional comments

Major comments:

1st. I believe that the author must discuss their data and connect it with the current discussion.

2nd. I am not convinced that this approach has a wider scale than coral gardening. I believe that the idea is the comparison between a sessile benthic group with a pelagic group. Despite the larval dispersion present of in both groups, the authors should state clearly in the text why they believe that this method has an action on a wider scale than others.

3rd. The authors begin their model separating the macroalgae from turf algae. However, in a later step, both are considered the same. These two groups have different effects on coral populations and on the reef ecosystem. There are several recent publications that report a phase shift to macroalgae as well as turf algae (Jouffray et al 2015, Naumann et al. 2015 and Gove et al 2015). Macroalgae are more aggressive in their interactions with coral populations; turf algae, in contrast, inhibit coral recruits, and also, in several cases, could be involved in the spread of coral disease. Furthermore, herbivorous fishes are more efficient at controlling turf algae than macroalgae. I recommend that the authors do separate model components for macroalgae dominance and turf algae dominance.

Minor comments:

1st. Line 15, “recovery of disturbed reefs is that of protection.” What does it mean?

2nd. Line 36, A period is missing in "2008) Nonetheless".

3rd. Line 68, A space is missing in “2012),stock”.

4th. Line 123, I do not agree with this assumption: “that all fishes will migrate from Reef I to Reef II.”

5th. Line 157, The definition of macroalgal phase shift as a stable state is controversial. There are many paper showing that there is no substantiated evidence to support this (Dudgeon et al. 2010, Zuychaluk et al. 2012). The persistence of a phase shift could be associated with another disturbance, such as sedimentation or increase of nutrients input. I therefore suggest avoiding the use of the term "stability".

6th. Line 196 to 198 and line 235 to 237. These recovery times are an output of this model. However there is insufficient precision this model to use this output as a real time of recovery. Many elements of this model were arbitrarily defined. Furthermore, as it is a general model, it could not be so precise. The recovery time should vary between different coral reef ecosystems. I suggest using relative rates of recovery. For example, where you now state that without intervention the model suggests 80 years to recovery while with intervention this time would be 50 years, you could say that according with this model, when the intervention are used, the recovery time is 62.5% of a system without intervention.

7th. Line 215 to 217. This is an assumption and should be moved to methods.

Reviewer 2 ·

Basic reporting

The manuscript “Potential contribution of fish restocking to the recovery of deteriorated reefs: an alternative restoration method?” by Obolski et al. presents a mathematical model to assess herbivores restocking as a restoration method for coral reefs. The study is relevant and the text is well written. However there are some issues that should be addressed before the manuscript publication. With this being said, I recommend major revisions.

Experimental design

Geographical contextualization is necessary. Can this methodology be applied to any coral reef from any location?

Details of what type/species of “grazers” are necessary. Authors should be more direct and specific or should define better what is the scope of the model.

Validity of the findings

Authors should stress the source of the fishes to restocking and the how this approach would affect (or not) the genetic diversity.

Authors should make clear to readers the model scope. They are working with coral reefs however in some parts the text authors refer to reefs, this can be confusing. Please add a clear explanation of possible location (ocean) of these reefs. This model is applicable to any kind of reef, from anywhere? If so, authors should explain and justify.

The model shows that the herbivores’ restocking increases coral cover or this is an assumption? This information should be clear in manuscript. Another useful information would be what type or species of corals (i.e. massive, branching, tabular) benefits from herbivores restocking. Notice that this consequences depends on the coral reefs location.

Additional comments

Abstract:
Lines 14-15: Sentence is not clear.

Lines17-19: Did some CRR have been successfully applied? Maybe a small introductory sentence would be placed to justify this sentence.

Line 20: Please specify what type (fish?) and/or specie of grazers

Line 22: This sentence would be improved including something like: “We study the effect of grazers fishes, as a restoration methods, in a mathematical…”.

Line 27: This is the first time stating that the study was carried out using herbivores fishes as a restoration method. This information should be in the beginning of the abstract (please see the comment above).

Introduction:
Line 40: Please check PeerJ citations guidelines. The double “))”looks odd. I could find some other examples in manuscript. Please check carefully.

Line 53: Authors propose grazer restocking as a additional CRR approach however is not clear what type or species of grazers? By the way a short sentence with the definition of fish grazers would be helpful in introduction. Is not clear also what type of reef (I suppose coral reef) this modeling approach is useful. What was the “reef model” was used to create the mathematical model? From Pacific, Caribbean? What geographical/taxonomic extent this approach can be used?

Line 62: Would you explicit some invertebrate examples?

Line 68: Add a space tab after citation.

Line 70: Please provide some examples of marine systems and locations.



Methods:

Maybe this section should be divided in more subsections. There are a lot of text in Results section that would be placed in Methods and Discussion sections (please see bellow).

Line 86: I feel “The model” very vague. Maybe authors may specify better.

Line 91: Please check citation guidelines.

Line 116: Migrates instead “migrating”? Please check previous sentences.

Lines 117-120: Maybe authors should include a short sentence explaining why they have estimated spillover rates using another methodology.

Line 123: I would recommend authors write Reef I and Reef II (Sentence case). This is a personal preference. I find easier to find and link to the “study locations”.

Lines 140-142: Sounds like an absolute truth. Please include a reference.

Results:
The second paragraph of this section would be placed in Methods section. I would recommend to revise results section to make it very objective and succinct.

Line 146: Results subtitles can be improved. I suggest including a full sentence with the main findings. “A single reef” is not informative enough.

Line 165: Please check citation guidelines.

Line 168: Please check citation guidelines.

Lines 172-180: Please make clear to readers how many fishes must be released per Km2 to restock be effective and how much it cost to restock a reef (hypothetically Reef I?).

Line 187-188: It is not clear in manuscript how would cost the herbivores restocking. “Relatively cheap” is too vague, add here the calculated value again.

Line 215: “relatively close” is vague, would authors estimate a distance?



Discussion:

Line 250: ad hoc (italic)?

Line 258: “We believe” sounds not “unscientific”, please change this sentence using evidences that this model/study provides. This can straighten your argument.

Lines 260-261: This sentence is very strong. Please provide some references.

Lines 262-265: This information should be also in introduction. However I have few questions regarding using this methodology:

1) What would be the consequences to genetic diversity?
2) What would be the ecological consequences (competition/homeostasis)?

Please add a short discussion/justification.

Line 266-268: This sentence is not clear enough. Please enumerate/summarize the “various situations”. Also, try to replace the word “help” to other more accurate.

Lines 269-271: What species already can be cultivated? Virtually “any kind” of herbivore can be? What are the herbivores species that can be cultivated in large scale? This information would justify the “low cost” of restocking? As far as I understood each fish would cost US$10. Please information about how many fishes are necessary per Km2 and how would totally cost to restock a reef (Reef I?). I suppose that US$10 for each fish is not so “cheap” for developing countries (where most of coral reefs are located).

Line 285: Macro algae eradication seems to be very expansive, how would it cost? Some reference?


Figures:

For all figures I would recommend not using red and green colors. Color blinded people won’t be able to distinguish the variables.

Fig 1 and 3– I would recommend to write “Coral” and “Macroalgae” in axis title. I also recommend adding the color scale units in figure. Line 411 have two “and”, please correct it.

Fig 2 and 4 – I would recommend to write the variables name in figures legend (i.e. Coral, Macroalgae and Herbivores) instead the abbreviations. “Variables” as Y axis title would be replaced to something more informative. Add a space tab between “Time” and “(years)” in X axis title. Authors would give an extra space among each plot, notice that “C” and “D” are very close from zeros.

---

## Round 0.2 · Minor Revisions

Please, include the explanations given in your letter in the revised version of your manuscript.

Reviewer 1 ·

Basic reporting

The manuscript 2015:09:6663:1:0:REVIEW entitled "Potential contribution of fish restocking to the recovery of deteriorated coral reefs: an alternative restoration method?" and written by Obolski et al. has improved since last version and now, it is ready to be published.

Experimental design

Experimental design is not apply to studies based on mathematical models.

Validity of the findings

As in all model approaches, giving the veracity of these model and theoretical assumptions, the results of this manuscript could help to point another way in restoration science on coral reefs.

Additional comments

Congratulation for this nice paper. I hope that it help to point another way to coral reef restoration.

Reviewer 2 ·

Basic reporting

Authors have improved significantly the manuscript however I have two concerns before publication:

1) Authors have included the following sentence in Methods section:

"The grazing fish graze macroalgae into algal turfs at a rate , where g(P) is taken to be P. Thus, the grazing fish reduce the rate of macroalgae growth over corals, while simultaneously expanding algal turfs, which can provide seabed for coral growth. Grazing fish are fished at rate f."

As far as I know expanding turf algae or "algal turfs" does not provides seabed for corals but the opposite. Many important articles shows that turf algae compete directly with corals for space or indirectly, stimulating pathogenic bacteria to grow on corals' tissue killing it. Please make these statements (regarding turf algae) very clear.

In response letter authors gave an explanation about the differences between "algal turfs" from "turf algae", I could not find these explanations in manuscript.

2) Authors have respond my comments about figures improvements. In the revised pdf I could not the improved figures. Please make sure to include the improved figures in the final version of the manuscript.

Best regards

Experimental design

No Comments

Validity of the findings

No Comments

Additional comments

No Comments

---

## Round 0.3 · accepted · Accept

Thank you for improving your manuscript as indicated.